# On the Design of an Energy Efficient Digital IIR A-Weighting Filter Using Approximate Multiplication

**DOI:** 10.3390/s21030732

**Published:** 2021-01-22

**Authors:** Ratko Pilipović, Vladimir Risojević, Patricio Bulić

**Affiliations:** 1Faculty of Computer and Information Science, University of Ljubljana, 1000 Ljubljana, Slovenia; ratko.pilipovic@fri.uni-lj.si; 2Faculty of Electrical Engineering, University of Banja Luka, 78000 Banja Luka, Bosnia and Herzegovina; vladimir.risojevic@etf.unibl.org

**Keywords:** band-pass digital filters, A-weighting filter, approximate computing, approximate filter, noise sensing

## Abstract

This paper presents a new A-weighting filter’s design and explores the potential of using approximate multiplication for low-power digital A-weighting filter implementation. It presents a thorough analysis of the effects of approximate multiplication, coefficient quantization, the order of first-order sections in the filter’s cascade, and zero-pole pairings on the frequency response of the digital A-weighting filter. The proposed A-weighting filter was implemented as a sixth-order IIR filter using approximate odd radix-4 multipliers. The proposed filter was synthesized (Verilog to GDS) using the Nangate45 cell library, and MATLAB simulations were performed to verify the designed filter’s magnitude response and performance. Synthesis results indicate that the proposed design achieves nearly 70% reduction in energy (power-delay product) with a negligible deviation of the frequency response from the floating-point implementation. Experiments on acoustic noise suggest that the proposed digital A-weighting filter can be deployed in environmental noise measurement applications without any notable performance degradation.

## 1. Introduction

Noise pollution is a common problem in urban environments. Humans are continuously exposed to noise as they go about their daily lives. However, exposure to noise in the urban environment or in the workplace can be a source of discomfort leading to health-related problems such as hearing loss if the correct protective actions are not taken. In order to assess the risk of noise for humans, one course of action is to measure the noise level present in the humans’ living and working environments. Typical environmental or background noise levels in residential areas range from 30 to 80 dB, and long-term exposure to sound levels over 85 dB causes hearing damage [1]. Studies [2,3] investigated the effects of exposure to office noise and showed that everyday exposure to noise disturbance affected comfort, health, and work performance.

In order to measure human exposure to noise, the measurement equipment must correlate the measured sound pressure level (SPL) to the perceived loudness level of noise using weighting filters, such as an A-weighting filter. An A-weighting filter for sound level meters is defined in the International standard IEC 61672-1 [4] and is required to assess noise levels for legislative regulations. The standard describes the A-weighting filter by tabulating frequency weighting values, and giving an analytical expression for the transfer function of the filter, but it does not define its implementation details. A-weighting is applied to the sound samples to estimate the loudness perceived by the human ear [1].

An A-weighting filter can be implemented as an analog or digital filter. Digital filters can achieve far superior results to those of analog filters, as they do not suffer from parasitic or temperature variations that affect analog filters. Besides, the implementation of digital filters in digital systems (e.g., SoCs and MCUs), which are omnipresent in today’s measurement equipment, sensor nodes, and edge devices, is straightforward.

Coefficients of digital filters obtained from transfer functions of analog filters are real numbers that require an infinite number of bits for their representations, or at least a floating-point representation in digital systems. In practical situations, it is impossible to represent a digital filter’s coefficients with an infinite number of bits; hence, designers generally use fixed-point approaches to represent the filter’s coefficients. Unfortunately, fixed-point designs degrade the filter frequency response and introduce a theoretical limit of the filter’s performance [5]. For example, in the realization of IIR filters in digital hardware, the filter’s accuracy is limited by the length of the word used to represent the coefficients and perform arithmetic operations. Additionally, due to quantization, these coefficients are not exact. Consequently, the finite-word filter’s frequency response with quantized coefficients is different from the filter’s frequency response with exact coefficients. On the other hand, a fixed-point digital filter design can maximize filter performance in terms of area, delay, and power consumption.

The interest in reducing the power consumption of digital filters used in edge computing and sensor networks is growing rapidly. Several techniques are used to achieve significant reductions in power consumption. One of the first papers proposing approximate processing to achieve these goals is [6]; there the authors proposed an algorithm to reduce the total switched capacitance by dynamically varying the filter order based on signal statistics. A recent trend in low-power design is approximate computing for reducing arithmetic activity and average chip power dissipation [7,8,9,10]. Multiplication represents a widespread arithmetic operation in DSP; therefore, many DSP applications can benefit from an efficient multiplier design. By relaxing the requirement for exact computation, we can design a power-efficient approximate multiplier [11,12,13,14,15]. The error that emerges from product approximation should be constrained to deliver acceptable results in the application. Therefore, it is essential to find a good compromise between the accuracy of a multiplier and its efficient design.

The effectiveness of approximate multipliers for achieving low-power processing motivated us to apply approximate multiplication inside the A-weighting filter. The main idea was to introduce approximate multiplication in an A-weighting IIR filter and save the area and energy while introducing a neglectable computational error. This work is motivated by some earlier works in which the approximate multiplication was used in finite impulse response (FIR) filters [16,17,18,19]. With the optimal placement of approximate multipliers inside the A-weighting filter, we anticipate that the frequency response would be almost identical to the frequency response of the A-weighting filter with exact arithmetic. In this work, we show that we can ensure the minimal influence of approximate multiplication on the performance of the A-weighting filter and achieve power-efficient processing.

The contributions of this paper can be summarized as follows:This paper presents a new design for an approximate low-power digital A-weighting filter implemented as a sixth-order IIR filter with approximate multipliers.This work provides a thorough analysis of the effects of approximate multiplication on the frequency response of an A-weighting IIR filter. We show how the optimal placement of approximate multipliers across the filter and the appropriate zero-pole pairings ensure minimal degradation of the filter’s frequency response in the presence of approximate multiplication.Synthesis results indicate that the proposed approximate IIR filter design achieves a nearly 70% reduction in energy (power-delay product) while preserving the required accuracy.

The rest of the manuscript is organized as follows. Section 2 gives some background, and discusses related work and the state of the art. The architecture of a digital IIR A-weighting filter and the effects of coefficient quantization are discussed in Section 3. The proposed approximate multiplication suitable for use in an IIR filter’s cascade is presented in Section 4. In Section 5, the impacts of zero-pole pairings and the placement of approximate multiplication among the filter’s characteristics are analyzed, followed by a description of the design of a low power digital A-weighting IIR filter using approximate multiplication. Experimental results are summarized in Section 6. Finally, the paper is concluded in Section 7.

## 2. Background and Related Work

### 2.1. Sound Level Measurement Basics

The human auditory system responds to air pressure changes, which are perceived as sound. Therefore, in order to quantify the sound level, it is convenient to measure the pressure of the sound wave at the location of the listener. The sound pressure level is computed as the root-mean-square (RMS) value of the sound pressure, $p_{RMS}$, relative to the reference pressure $p_{0}=20\;\mu \mathrm{Pa}$ and expressed in decibels [20].
(1)$$SPL=20\log\frac{p_{RMS}}{p_{0}}\;\;\;\left[\mathrm{dB}\right].$$

Reference value $p_{0}$ is chosen to be approximately the threshold of hearing at 1000 Hz, for a typical human ear. The effective sound pressure is the RMS value of the instantaneous sound pressure *p* over a given interval of time. The RMS value of the sound pressure is defined as
(2)$$p_{RMS}=\sqrt{\frac{1}{T}\int_{0}^{T}p^{2}\left(\tau \right)d\tau }.$$

Since the root mean square computation in Equation (Equation 2) involves time averaging, three values for the time constant *T* are adopted in sound level measurements, namely, impulse (I), fast (F), and slow (S) averaging with time constants equal to 35, 125, and 1 s, respectively [20].

### 2.2. A-Weigthing Filter

The human auditory system has a more pronounced response to signals in the frequency range between 500 and 8 kHz and is less sensitive to very low-pitch and high-pitch noises. To ensure that a sound level meter measures close to what a human hears, the correct frequency weighting related to the response of the human auditory system must be used in sound level measurement. The A-weighting filter [4] is designed with this goal in mind and subsequently has become the most commonly used frequency response in sound level meters. Despite its shortcomings, in many countries, the use of the A-weighting frequency filter is mandatory for the measurement of environmental and industrial noise and assessments of potential hearing damage and health effects of noise.

The A-weighting filter, whose magnitude response is presented in Figure 1, is a bandpass filter designed to simulate the perceived loudness of low-level tones. It progressively de-emphasizes frequencies below 1000 Hz. At 1000 Hz, the filter gain is 0 dB. Between 1000 and about 5000 Hz the signal is slightly amplified, and at about 5000 Hz and higher, the signal is attenuated. The transfer function of an analog A-weighting filter is defined in [4] as:(3)$$H_{a}\left(s\right)=\frac{4\cdot \pi^{2}\cdot 12194^{2}\cdot s^{4}}{{\left(s+2\pi \cdot 20.6\right)}^{2}{\left(s+2\pi \cdot 12194\right)}^{2}\left(s+2\pi \cdot 107.7)(s+2\pi \cdot 739.9\right)}$$

### 2.3. A-Weighting Filter Design

Most of the previous work on noise measurement [21,22,23,24,25,26,27,28], has been done using the analog A-weighting filter defined by (Equation 3). Usually, such a filter consists of several active stages implemented with operational amplifiers. Hakala et al. [21] and Kivelä et al. [22,23] presented a sensor node for acoustic noise measurement which uses an analog A-weighting filter. They claim that a digital filter with real-valued coefficients involves excessive floating-point calculations, which surpasses the limit of a small, off-the-shelf integer-based MCU. Consequently, they implemented an A-weighting filter with a cascade of three analog high-pass filters and two analog low-pass filters. The paper by Rimell et al. [1] describes the implementation of the weighting filters as digital IIR filters. It provides all the necessary formulae to calculate the filter coefficients for any sampling frequency directly. The authors used a bilinear transformation to transform the analog equations that are provided in [4]. The downside of using a bilinear transform to convert an analog filter to a digital one is that the transfer function of a digital filter does not strictly follow the analog frequency response at higher frequencies. Risojević et al. [29] proposed a sensor node capable of sound level measurement based on a hardware platform with limited computational resources. Furthermore, to reduce the communication between the sensor node and a sink node and the power consumed by the IEEE 802.15.4 (ZigBee) transceiver, they performed digital A-weighting filtering on the node. The proposed digital A-weighting filter’s coefficients were obtained using a matched-z transformation, and the filter was implemented as a cascade of three second-order IIR sections with quantized coefficients. In contrast to [1], Risojević et al. added a low-pass section for correction of the magnitude response at higher frequencies. In such a way, they obtained a digital filter that satisfies the tolerance limits imposed by the IEC 61672-1 standard.

### 2.4. Approximate Digital Filters

Many DSP applications use distributed arithmetic based approximate structures for efficient implementation of inner products. In the existing literature, most of these approximate architectures are developed by truncating the least significant bits (LSBs) of the inputs or filter coefficients [16,18,19,30,31]. As FIR filters are more tolerant towards computational errors than IIR filters, many attempts to avoid costly multiplications in FIR filters using distributed arithmetic structures have been made in the last four decades [16,17,18,19]. On the other hand, to the best of our knowledge, no attempts have been made to implement IIR filters using approximate arithmetics with coefficient quantization and finite word-length. What follows is an overview of the most recent related work on approximate filter design.

The paper [32] tries to reduce the number of adders of the multiplier block to reduce overall chip area and power consumption. It proposes a power-oriented optimization method for linear phase FIR filters. In the proposed algorithm, the average adder depth of the structural adders is used as the optimization objective in the discrete coefficients search. The authors showed that power savings could be as much as 19.6%. Kumm et al. [17] presented two novel optimization methods based on integer linear programming that minimize the number of adders used to implement a direct/transposed FIR filter. The proposed algorithms work by bounding the adder depth used for these products, which can be used to design filters for low power applications. In contrast to previous multiplier-less FIR approaches, the methods introduced in Kumm et al. [17] ensure optimal adder count. In [16], the authors proposed a fixed-point adaptive FIR filter using approximate distributed arithmetic circuits. The radix-8 Booth algorithm was used to reduce the number of partial products. Additionally, the partial products were approximately generated by truncating the input data. The proposed adaptive FIR filter was employed to identify an unknown system. The authors considered 64-tap and 128-tap FIR adaptive filters to assess the proposed design as low and high order applications. Synthesis results showed that the proposed design achieves, on average, a 55% reduction in energy.

Volkova et al. [33] proposed a generic methodology for the construction of IIR filters that behave as if the computation was performed with infinite accuracy, then converted to the low-precision output format with an error smaller than its least significant bit. This generic methodology is detailed for low-precision IIR filters in the Direct Form I implemented in FPGA logic. The authors validated the proposed methodology on a range of IIR filters. In the paper [34], an IIR filter’s hardware complexity is iteratively reduced by approximating the IIR filter coefficients to maximize the number of eliminable common subexpressions. The authors showed that by using the proposed algorithm, a high-order lowpass filter with a minimum stopband attenuation of 60dB could be implemented by a 13-tap IIR filter with a group delay deviation of 0.002 only. Logic synthesis showed that the proposed IIR design saves 39.4% of the area and 41.8% of power consumption over the FIR solutions. The work in [35] proposes an IIR filter implementation considering the quantization aspect. The authors have proposed a pipelined IIR filter structure and a novel implementation of the quantizer. Finally, the work in [36] proposes fixed-point hardware architectures for IIR filters, focusing on design specifications for ECG signal processing, using the truncation error feedback to attenuate errors caused by finite word length operations inside IIR recursive structures. The proposed IIR filter architectures were described and simulated using Verilog and synthesized using the 45 nm Nangate Open Cell Library to verify the area, delay, and power metrics.

However, there is no thorough analysis of the effect of approximate multiplication, quantization, and zero-pole pairings in the IIR digital filters, as we show in this work.

## 3. Digital IIR A-Weighting Filter Architecture and Coefficient Quantization

A digital A-weighting filter is implemented as infinite impulse response (IIR) filter, whose output depends on a finite number of input samples and a finite number of previous filter outputs. Due to the feedback paths, IIR filters are less numerically stable than their FIR counterparts [37] but provide better performance and less computational cost than FIR filters. In this section, we explore a suitable implementation of a digital A-weighting filter and its coefficient quantization.

We follow the approach by Risojević et al. [29]. Using matched-z transformation [37] for the transfer function given in (Equation 3) of the analog A-weighting filter, and sampling frequency $F_{S}=48$ kHz, the transfer function of the A-weighting digital filter is obtained as:(4)$$H_{d}\left(z\right)=\frac{{\left(1-z^{-1}\right)}^{4}}{{\left(1-0.9973z^{-1}\right)}^{2}{\left(1-0.2025z^{-1}\right)}^{2}\left(1-0.9860z^{-1}\right)\left(1-0.9097z^{-1}\right)}$$

The magnitude response of the filter with transfer Function (Equation 4) slightly violates the tolerance limits imposed by [4] for high frequencies. Therefore, we added a first-order low-pass section to correct the magnitude response. The gain and cutoff frequency of the added first-order section were chosen by trial and error. The resulting transfer function is:(5)$$H\left(z\right)=\frac{\left(1+0.3z^{-1}\right){\left(1-z^{-1}\right)}^{4}}{{\left(1-0.9973z^{-1}\right)}^{2}{\left(1-0.2025z^{-1}\right)}^{2}\left(1-0.9860z^{-1}\right)\left(1-0.9097z^{-1}\right)}$$

The digital filter defined by (Equation 5) will be referred as a reference filter in the rest of the paper.

As can be seen from (Equation 5), the filter has poles in the unit circle’s proximity, which can make the filter unstable in the presence of coefficient quantization. Risojević et al. [29] employed a cascade-form realization of the transfer function given in (Equation 5) using second-order sections (SOS) to avoid system instability due to the round-off errors in the fixed-point arithmetic. The main disadvantage of SOS filter implementation is the nonlinear relationship between the filter’s coefficients and filter’s poles and zeros [37]. Due to this nonlinear relationship, it is hard to determine the effect of quantization of the filter coefficients on its poles and zeros’ positions and control the sensitivity of these positions to quantization errors. The SOS’s nonlinear relationship between coefficients and poles motivated us to redesign the A-weighting filter as a cascade-form with the first-order sections (FOS). The filter implementation using FOS is characterized by a linear relationship between filter coefficients and its zeros and poles. Hence, we have control of the poles and zeros of the filter with quantized coefficients. Moreover, the A-weighting filter’s FOS and SOS implementations have the same number of employed delay elements and arithmetic units (adders and multipliers). Factorization of the numerator and denominator polynomials in the transfer function of the A-weighting digital filter (Equation 5) yields the cascade-form implementation with FOS:(6)$$H\left(z\right)=H_{1}\left(z\right)H_{2}\left(z\right)H_{3}\left(z\right)H_{4}\left(z\right)H_{5}\left(z\right)H_{6}\left(z\right)\;,$$
where the transfer functions of the first-order sections are:(7)$$\begin{array}{cc} H_{1}\left(z\right) & =\frac{1}{1-0.2025z^{-1}}\\ H_{2}\left(z\right) & =\frac{1+0.3000z^{-1}}{1-0.2025z^{-1}}\\ H_{3}\left(z\right) & =\frac{1-1.0000z^{-1}}{1-0.9860z^{-1}} \end{array}\;\begin{array}{cc} H_{4}\left(z\right) & =\frac{1-1.0000z^{-1}}{1-0.9079z^{-1}}\\ H_{5}\left(z\right) & =\frac{1-1.0000z^{-1}}{1-0.9973z^{-1}}\\ H_{6}\left(z\right) & =\frac{1-1.0000z^{-1}}{1-0.9973z^{-1}}\;, \end{array}$$

The proposed filter can also be represented by matrices of its coefficents as:(8)$$B=\left[\begin{array}{cc} 1.0000 & 0\\ 1.0000 & \;\;\;\;0.3000\\ 1.0000 & -1.0000\\ 1.0000 & -1.0000\\ 1.0000 & -1.0000\\ 1.0000 & -1.0000 \end{array}\right]\;A=\left[\begin{array}{cc} 1.0000 & -0.2025\\ 1.0000 & -0.2025\\ 1.0000 & -0.9860\\ 1.0000 & -0.9079\\ 1.0000 & -0.9973\\ 1.0000 & -0.9973 \end{array}\right]\;,$$
where the position of the coefficients (i.e., zeros and poles) within the matrices represents the placement of FOS. Cascade filter realizations can be obtained by different pole-zero pairings and by different orderings of sections. In floating-point arithmetic, pole-zero pairings and the order of sections in the cascade do not affect the filter’s frequency response. However, when the filter is applied in digital electronics using the finite number of bits to represent the filter’s coefficients and in the presence of approximate multiplication, we cannot presume that the filter’s frequency response is unaffected by pole-zero pairings, the ordering of FOS in the cascade and approximate arithmetics.

We tackle this problem in Section 5. Here, we present the proposed quantization used to determine the minimal amount of bits required to represent the filter coefficients without violating the tolerance limits imposed by the IEC 61672-1 standard. We perform quantization as follows:(9)$$\beta_{iq}=\frac{\mathrm{round}(\beta_{i}\cdot 2^{Q})}{2^{Q}}\;,\;\alpha_{iq}=\frac{\mathrm{round}(\alpha_{i}\cdot 2^{Q})}{2^{Q}}\;,$$
where round() represents rounding to the nearest integer, $\beta_{iq}$ and $\alpha_{iq}$ represent quantized coefficients obtained from $\alpha_{i}$ and $\beta_{i}$, and *Q* denotes the number of bits used to represent the decimal part of the filter coefficients.

The magnitude responses of the A-weighting filter for different values of *Q* are depicted in Figure 2. When Q=8, the filter’s frequency response violates the IEC 61672-1 standard’s tolerance limits, but only in a narrow frequency range from 10 to 100 Hz. For Q=9, the filter’s magnitude response has a 0.3 dB higher magnitude response than upper tolerance limits for frequencies smaller than 20 Hz. Finally, an A-weighting filter with Q=10 satisfies the tolerance limits imposed by the IEC 61672-1 standard. Therefore, we represent the coefficients with 11 bits in the two’s complement fixed-point format. The quantized filter coefficients for all six FOS of the A-weighting filter multiplied by $2^{10}$ are:(10)$$B=\left[\begin{array}{cc} 1024 & 0\\ 1024 & 307\\ 1024 & -1024\\ 1024 & -1024\\ 1024 & -1024\\ 1024 & -1024 \end{array}\right]\;A=\left[\begin{array}{cc} 1024 & -207\\ 1024 & -207\\ 1024 & -1010\\ 1024 & -932\\ 1024 & -1021\\ 1024 & -1021 \end{array}\right]\;.$$

## 4. The Proposed Approximate Multiplication

In digital filters, the multipliers represent indispensable components that have a strong influence on their area, delay, and energy. If we employ approximate multiplier in digital filters, we can significantly improve energy consumption and area usage. Low energy consumption is a desired property as A-weighting filters are often employed as a part of battery-powered devices. However, approximate multiplication can significantly influence the A-weighting filter’s stability and magnitude response. Hence, careful design and placement of approximate multipliers are required. This section first presents an exact multiplier whose design leverages the coefficient’s quantization and then proposes an approximate multiplier, which we obtain by simplifying the exact multiplier.

### 4.1. Exact Radix-4 Multiplier

A radix-4 Booth multiplier [38] consists of two stages: a partial product generation, and a partial product addition stage. Let us illustrate radix-4 Booth encoding for the multiplication of two *n*-bit integers, i.e., a multiplicand *X* and multiplier *Y* in two’s complement:(11)$$\begin{array}{c} X=-x_{n-1}\cdot 2^{n-1}+\sum_{i=0}^{n-2}x_{i}\cdot 2^{i}\;, \end{array}$$
and
(12)$$\begin{array}{c} Y=-y_{n-1}\cdot 2^{n-1}+\sum_{j=0}^{n-2}y_{j}\cdot 2^{j}\;, \end{array}$$
where $x_{i}$ and $y_{j}$ represent the bits from *X* and *Y*, respectively. In the radix-4 Booth encoding, the multiplier *Y* is divided into overlapping groups of three bits:(13)$$\begin{array}{ccc} Y & = & \sum_{j=0}^{\left\lceil n/2\right\rceil -1}{\hat{y}}_{j}^{R4}\cdot 4^{j}\;, \end{array}$$
where
(14)$${\hat{y}}_{j}^{R4}=-2y_{2j+1}+y_{2j}+y_{2j-1},\;\;{\hat{y}}_{j}^{R4}\in \left\{0,\pm 1,\pm 2\right\}\;.$$

Taking into account the radix-4 enconding of *Y*, we can write the product P=X·Y as:(15)$$P=\sum_{j=0}^{\left\lceil n/2\right\rceil -1}PP_{j}\cdot 4^{j}\;,$$
where $PP_{j}$ represents *j*-th partial product generated from ${\hat{y}}_{j}^{R4}$ group encoding:(16)$$PP_{j}=X\cdot {\hat{y}}_{j}^{R4},\;\;PP_{j}\in \left\{0,\pm X,\pm 2X\right\}\;.$$

The previous discussion deals with the general case of an *n*-bit multiplier. In our case, the filter coefficients of the A-weighting filter are represented with 11-bit integers. If we observe filter coefficients as Y input, the resulting radix-4 Booth multiplier generates six partial products, as shown in Figure 3a). As we can see, the partial product generation stage consists of six Booth encoders, which generate partial products from each radix-4 group ${\hat{y}}_{j}^{R4}$. In the partial product addition stage, we employ the Wallace tree [39] to reduce the number of partial products to two. The final partial product addition is implemented using a prefix (fast) adder [38].

### 4.2. Approximate Odd Radix-4 Multiplier

In order to reduce the number of partial products, we propose a slightly modified radix-4 encoding. The main idea behind the proposed encoding is to shift the position of group encodings one place to left, as illustrated in Figure 3b. Let:(17)$${\tilde{y}}_{j}^{R4}=-2y_{2j+2}+y_{2j+1}+y_{2j}$$

Now, the encoded value, $Y_{ODD}$, of an *n*-bit binary number is equal to:(18)$$Y_{ODD}=\sum_{j=0}^{\left\lceil n/2\right\rceil -1}{\tilde{y}}_{j}^{R4}\cdot 4^{j}$$

In the case of an 11-bit number, the encoded value $Y_{ODD}$ is:(19)$$\begin{array}{ccc} Y_{ODD} & = & \sum_{j=0}^{4}{\tilde{y}}_{j}^{R4}\cdot 4^{j}=\left(-2y_{10}+y_{9}+y_{8}\right)4^{4}+\ldots +\left(-2y_{2}+y_{1}+y_{0}\right)4^{0}\\ & = & -y_{10}2^{9}+y_{9}2^{8}+\ldots +y_{2}2^{1}+y_{1}2^{0}+y_{0}2^{0}\;. \end{array}$$

By setting $y_{0}2^{0}=2y_{0}2^{-1}$, we can rewrite the above equation as:(20)$$\begin{array}{ccc} Y_{ODD} & = & -y_{10}2^{9}+y_{9}2^{8}+\ldots +y_{2}2^{1}+y_{1}2^{0}+y_{0}2^{-1}+y_{0}2^{-1}\\ & = & \left(-y_{10}2^{10}+\sum_{i=0}^{9}y_{j}2^{j}\right)/2+y_{0}/2\;. \end{array}$$

Hence, the idea is to use the encoding from (Equation 17) and (Equation 19) to encode the multiplier Y:(21)$$\begin{array}{ccc} Y & = & 2\sum_{j=0}^{4}{\tilde{y}}_{j}^{R4}\cdot 4^{j}-y_{0}=2Y_{ODD}-y_{0}\;. \end{array}$$

In such a way, we can decrease the number of partial products by one for binary numbers with odd number of bits.

To avoid costly subtraction, which leads to a more complex circuitry, we propose to neglect the term $y_{0}$ and to approximate *Y* as follows
(22)$$\begin{array}{c} Y\approx \hat{Y}=2\sum_{j=0}^{4}{\tilde{y}}_{j}^{R4}\cdot 4^{j}\;, \end{array}$$

Section 5 shows that neglecting the term $y_{0}$ leads to an acceptable error. From (Equation 22), we can see that an error arises only when *Y* is an odd number.

With the proposed approximate odd radix-4 encoding, we can calculate the product $P\approx X\cdot Y_{ODD}$ as:(23)$$P\approx X\cdot \hat{Y}=2\sum_{j=0}^{4}{\tilde{PP}}_{j}\cdot 4^{j}\;,$$
where ${\tilde{PP}}_{j}=X\cdot {\tilde{y}}_{j}^{R4}$ represents *j*-th partial product generated from ${\tilde{y}}_{j}^{R4}$. Note that we employ the same circuitry to obtain ${\tilde{y}}_{j}^{R4}$ and ${\tilde{PP}}_{j}$ as in the design of exact radix-4 multiplier.

To further improve the proposed multiplier design in terms of area, delay, and energy consumption, we propose the omission of the last *M* bits of multiplier *Y*. The proposed omission also decreases the number of partial products, leading to even more hardware and energy-efficient design. For example, if M=5, we omit the last two partial products in Figure 3b). Section 5 shows that this error does not affect the filter’s response if we select *M* carefully in each first-order section.

### 4.3. Error Analysis of the Approximate Odd Radix-4 Multiplier

In this subsection, we present the error analysis of the approximate odd radix-4 (AO-RAD4) multiplier presented in the previous subsection. We analyze the mean relative error (MRE) and the relative error distribution for error assessment. MRE is obtained as an average relative error for all sets of inputs and all possible combinations for a n×11 bit multiplier.

The calculation of relative error for AO-RAD4 is as follows. Considering (Equation 22) and (Equation 23), the relative error of AO-RAD4 multiplier for a number pair (X,Y) is obtained as:(24)$$RE\left(X,Y\right)=\frac{|X\cdot \hat{Y}-X\cdot Y|}{\left|X\cdot Y\right|}=\frac{|\hat{Y}-\;Y|}{\left|Y\right|}\;,$$
where $\hat{Y}$ is an approximately encoded operand as in (Equation 22). Hence, the relative error depends only on *Y*. The mean relative error (MRE) is calculated as follows:(25)$$\begin{array}{ccc} MRE(X,Y) & = & \frac{1}{2^{11}}\sum_{Y}RE\left(X,Y\right)=\frac{1}{2^{11}}\sum_{Y=-2^{10}}^{2^{10}-1}\frac{|\hat{Y}-\;Y|}{\left|Y\right|}\;. \end{array}$$

Figure 4 illustrates MRE (left) and error distribution (right) for different design instances of the AO-RAD4 multiplier. Error distribution is the probability that the relative error is smaller than a specific value. We can notice that MRE (Figure 4, left) increases exponentially with *M*. The error distribution (Figure 4, right) shows that the parameter *M* has a significant impact on error distribution. For example, the number of outputs whose relative error is below 0.1 decreases significantly (from 93% to 86%) when the parameter *M* increases from 3 to 4.

## 5. Hardware Implementation of the Digital A-Weighting Filter with Approximate Multiplication

In this section, we assess the influence of the placement of approximate multipliers inside the digital A-weighting filter and the influence of the zero-pole pairing and ordering of FOS within the digital filter.

### 5.1. Influence of Approximate Multipliers Placement on the Frequency Response

Employment of approximate multiplication in the A-weighting filter requires careful placement of approximate multipliers across the FOS cascade. The simple substitution of exact multipliers with approximate ones can lead to violation of the filter’s requirements or even make the system unstable.

To determine the optimal placement of the AO-RAD4 approximate multipliers within the digital A-weighting filter, we evaluated the magnitude response of the digital A-weighting filter in the presence of approximate multiplication. For every coefficient, we replaced the exact radix-4 multiplier with different instances of AO-RAD4 multiplier while keeping other multipliers exact. Then, we checked whether the proposed digital filter’s magnitude response satisfies the criteria for the A-weighting filter. Moreover, we quantitatively assessed the similarity between magnitude responses of the proposed and the reference A-weighting filter, given by (Equation 5) using the cross signature scale factor (CSF) [40,41]. The CSF factor is used to quantify the amplitude difference between frequency responses. For a specific frequency $\omega_{k}$, CSF is defined as:(26)$$\begin{array}{cc} CSF\left(\omega_{k}\right)= & \frac{2\left|H^{*}\left(\omega_{k}\right)H_{R}\left(\omega_{k}\right)\right|}{{\left|H\left(\omega_{k}\right)\right|}^{2}+{\left|H_{R}\left(\omega_{k}\right)\right|}^{2}},\;k=1,2,\ldots ,N\;, \end{array}$$
where $H_{R}\left(\omega_{k}\right)$ and $H(\omega_{k})$, represent the reference and the proposed frequency responses at frequency $\omega_{k}$, respectively, and *N* represents number of frequency points. The CSF ranges from 0 to 1.

Table 1 reports mean CSF for different values of the parameter *M* when AO-RAD4 multiplier is applied to different coefficients (factors). The combinations under which examined A-weighting filter satisfies tolerance limits for the frequency response are marked in green; otherwise, they are marked in red. As expected, the multiplications with coefficients in the FOS whose poles and zeros are further from the unit circle, are more tolerant to approximation errors and can have larger *M*. Now, the multiplication with the coefficients from (Equation 10) is as follows. As can be observed from Table 1, multiplication with factors 207 and 307 could be replaced with AO-RAD4 with truncation parameter M=6. However, AO-RAD4 multipliers with M=5 and M=6 have the same number of partial products. Hence, it is better to use AO-RAD4 with M=5 as it has significantly better MRE. When multiplying with 932, AO-RAD4 with M=5 can be used. When multiplying with 1010, AO-RAD4 with M=4 can be used. We can also see that multiplication with 1021 is very sensitive to approximation error, and we cannot use the AO-RAD4 multiplier. Hence, we employ the exact radix-4 multiplier for multiplication with 1021.

### 5.2. Influence of FOS Placement on the Frequency Response

In floating-point arithmetic, the position of sections in a cascade does not affect the filter’s impulse response. However, we used fixed-point arithmetic combined with approximate multiplication, so we cannot presume that impulse response is unaffected by FOS’s position in the cascade. To find the optimal zero-pole pairings and FOS placement, we evaluated all possible combinations of zeros and poles and the position of FOS in the cascade. We have calculated the frequency response for every combination and compared it to the reference A-weighting filter frequency response using CSF measure. The evaluation revealed that the following FOS cascade achieves the best CSF:(27)$$B=\left[\begin{array}{cc} 1024 & 307\\ 1024 & -1024\\ 1024 & -1024\\ 1024 & -1024\\ 1024 & -1024\\ 1024 & 0 \end{array}\right]\;A=\left[\begin{array}{cc} 1024 & -932\\ 1024 & -1021\\ 1024 & -1021\\ 1024 & -1010\\ 1024 & -207\\ 1024 & -207 \end{array}\right]$$

Finally, the proposed digital multiplier with the optimal placement of AO-RAD4 multipliers and the optimal pairings and order of FOS is presented in Figure 5.

### 5.3. The Stability of Proposed Filter

In terms of poles and zeros, a digital filter is stable if and only if all poles of the filter’s transfer function reside inside the unit circle in the *z*-plane. Two poles that correspond to coefficients with the value −1021 (Equation 27) are unaffected by approximate multiplication, as the proposed filter employs exact multiplication for these coefficients. To determine the influence of approximate multiplication on the remaining poles, we should first analyze the effects of product approximation on the coefficients. The approximate multiplication alters operand *Y*, and the operand *X* remains unchanged (see (Equation 22) and(Equation 23)). The $\hat{Y}$ in (Equation 22) is always smaller than *Y*, so the approximate product is always smaller than the exact product. When we apply approximate multiplication in the filter, we select the filter’s coefficients as operand *Y*. As we perform the exact addition, the computational error solely depends on the multiplication. The approximate multiplication leads to a decrease of coefficients, which decreases the absolute pole values and moves poles away from the unit circle. Therefore, the proposed approximate multiplication cannot lead to an unstable filter.

In addition to pole analysis, we have also evaluated the impulse response of the proposed filter. We have calculated the upper and lower impulse response envelopes using the Hilbert-transform FIR filter [42]. We chose the Hilbert-transform FIR filter to calculate the envelopes because it produces the most accurate envelope estimation. Figure 6 depicts the impulse response of the proposed filter, together with the envelopes for the first 50 samples. As we can see, both envelopes and the impulse response h(n) rapidly decay to zero. From the standpoint of the impulse response, we can conclude that the proposed filter is stable.

## 6. Simulation and Synthesis Results

We performed the experiments in three steps to verify the proposed approach for implementing an IIR A-weighting filter. Firstly, MATLAB simulations are described and presented to assess the fixed-point A-weighting IIR filter’s behavior with and without approximate multiplication. MATLAB simulation consists of comparing the frequency responses of the filters, filtering a set of environmental noise recordings, and comparing the filters’ outputs in terms of normalized root mean square error (NRMSE) and mean absolute error of sound pressure level (SPL). Secondly, we have used Verilog to implement the filters and synthesize them to 45 nm Nangate Open Cell Library. The resulting values of the area, delay, and power performance are reported. Finally, we have implemented the filter in Zynq-7000 SoC on the ZYBO Z7 FPGA development board to verify the filter’s operation in a real environment.

### 6.1. Magnitude Response of the Proposed Digital A-Weighting Filter

In this section, we present the MATLAB simulations of the proposed and reference A-weighting filters to observe the influence of approximate multiplications on the frequency response of the filter. We observe how much the frequency response of the proposed filter deviates from the exact frequency response given in the standard.

Figure 7 shows the magnitude responses of the proposed digital A-weighting filter from Figure 5 and the reference digital A-weighting filter whose transfer function is given by (Equation 5). Note that in MATLAB simulation, we use IEEE754 double-precision format to represent the reference filter’s coefficients. It can be observed from Figure 7 that the magnitude response of the proposed A-weighting filter satisfies the tolerance limits imposed by IEC 61672-1 standard. Moreover, the magnitude responses of the proposed and reference digital A-weighting filters are almost identical to each other.

To quantitatively assess the two magnitude responses, we used the CSF measure. Figure 8 shows CSF for the frequency range $\left[10\;\mathrm{Hz},20\;\mathrm{kHz}\right]$. The high values of CSF for the examined frequency range suggest that the implemented and reference A-filter have nearly identical frequency responses. For the examined frequency range, the average CSF equals 99.43 %, which indicates a high similarity between frequency responses of the reference and the proposed A-weighting filters. Therefore, we can conclude that employed approximate multipliers have a negligible influence on the filter’s frequency response.

### 6.2. Acoustic Noise Level Measurement

To assess the proposed A-weighting digital filter’s performance with approximate multipliers, we used the DEMAND collection of acoustic noise in diverse environments [43,44]. For acoustic noise level measurement, we have calculated each recording’s sound pressure level according to Equation (Equation 1) using fast averaging. Each recording is frequency A-weighted before we calculate the SPL value to take into account the impact of frequency on human perception of loudness. The DEMAND collection of recordings comprises four indoor environments categories, with three recordings within each category. The indoor categories are Domestic, Office, Public, and Transportation. The Domestic category consists of DKITCHEN (inside a kitchen during the preparation of food), DLIVINGR (inside a living room), and DWASHING (domestic washroom with washing machine running) recordings. The Office category consists of OHALLWAY (a hallway inside an office building with occasional traffic), OMEETING (a meeting room), and OOFFICE (a small office with three people using computers) recordings. The Public category consists of PCAFETER (a busy office cafeteria), PRESTO (a university restaurant at lunchtime), and PSTATION (the main transfer area of a busy subway station) recordings. Finally, the Transportation category consists of the following recordings: TBUS (a public transit bus), TCAR (a private passenger vehicle), and TMETRO (a subway).

Figure 9 shows the normalized root mean square error (NRMSE) between the signal from the reference filter and the signal from the proposed filter for each of the recordings in the DEMAND collection. Normalized root mean square error is defined as:(28)$$NRMSE=\frac{\sqrt{\frac{1}{N}\sum_{n=0}^{N-1}{\left(x_{r}\left[n\right]-x_{a}\left[n\right]\right)}^{2}}}{\left(x_{r,max}-x_{r,min}\right)},$$
where $x_{r}$ is the signal obtained from the reference digital filter, $x_{a}$ is the signal obtained from the proposed filter, $x_{r,max}$ and $x_{r,min}$ are the maximum and minimum values of the signal $x_{r}$, respectively, and *N* is the number of samples in each signal. It can be observed from Figure 9 that the NRMSE values between the signal from the reference filter and the signal from the proposed filter are very small. To statistically assess the range of estimates for mean NMRSE, we have calculated a 95% confidence interval (95% CI) from the obtained NMRSE on the DEMAND dataset. The CI determines the range of plausible values for mean NMRSE. The CI is calculated as follows:(29)$$\hat{X}\pm t_{c}\left(\frac{s}{\sqrt{n}}\right)\;,$$
where $\hat{X}$ represents the mean value of observed samples, $t_{c}$ represents the critical value from the Student’s t-distribution, *s* represents the standard deviation of observed samples, and *n* represents the number of samples. We have obtained 95% CI of $\left(26.85\pm 11.28\right)\cdot 10^{-4}$ for the estimate of mean NMRSE. Hence, our method would exhibit NMRSE between $15.57\cdot 10^{-4}$ and $38.13\cdot 10^{-4}$, which implies that the proposed filter can be deployed in sound pressure level measurement without noticeable performance degradation.

We have calculated two sound pressure levels for each recording: one with the proposed and one with the reference A-weighting filter. The loudness was calculated using the “fast” response (window size of 250 ms). The mean error (${\bar{\Delta }}_{SPL}$) is also reported for each recording. Figure 10 shows the loudness profiles for each of the recordings in dB SPL (A-weighted).

As can be observed from Figure 9 and Figure 10, NRMSE between the signal from the reference filter and the signal from the proposed filter is in strong correlation with the mean absolute error (${\bar{\Delta }}_{SPL}$) between the SPL values obtained with the proposed and the reference A-weighting filters. For example, the DKITCHEN recording has the smallest NRMSE and ${\bar{\Delta }}_{SPL}$, and the TCAR recording has the highest NRMSE and ${\bar{\Delta }}_{SPL}$.

To understand the underlying distribution of $\Delta_{SPL}$, we have calculated the histogram for the DEMAND dataset and presented it in Figure 11. From Figure 11, we can conclude that a significant amount of the $\Delta_{SPL}$ concentrates on interval [0.6,0.8] dB. Through the histogram analysis, we concluded that 91% percent of obtained $\Delta_{SPL}$ is smaller than 1 dB. Keeping in mind that professional SPL meters tend to have ±1 dB error tolerance, these results indicate that the proposed filter offers satisfiable performance for SPL measurement. Finally, we can see that the maximal $\Delta_{SPL}$ is equal to 1.4 dB. This suggests that the proposed filter can comply with an Type 2 sound level meter [4].

Finally, we have assessed the proposed filter’s decibel range using pink noise sequences. We have generated several pink noises with different noise levels and calculated two sound pressure levels for each sequence: one with the proposed approximate and one with the reference A-weighting filter. Figure 12 shows the correlation between the noise level of pink noise and ${\bar{\Delta }}_{SPL}$. As we can see, the proposed filter gives satisfactory results for the examined pink noise sequences.

### 6.3. CMOS Synthesis

In this subsection, we analyze and compare the proposed digital A-weighting filter’s hardware performance in terms of power, area, delay, and power-delay-product (PDP). We compare the synthesis results of two digital A-weighting filters: the proposed digital filter with AO-RAD4 multipliers as in Figure 5, and the reference filter with exact RAD-4 multipliers (Equation 5). The filters were implemented in Verilog and synthesized to 45 nm Nangate Open Cell Library. For Verilog to GDS synthesis flow, we employed OpenROAD Flow [45], a full RTL-to-GDS flow built entirely on open-source tools. We used timing with 10 MHz virtual clocks to evaluate the power with a 5% signal toggle rate and output load capacitance equal to 10 fF. The synthesis conditions aim to compare different filters while keeping equal conditions for all experiments. The synthesis results are listed in Table 2 and consist of cell area in $\mu m^{2}$, delay or critical path in nanoseconds, total power (leakage plus dynamic) in μW, and energy or power-delay-product (PDP) in fWs.

As can be observed from Table 2, the proposed digital filter with the approximate AO-RAD4 has substantially smaller area utilization and energy consumption compared to the reference digital filter with exact RAD-4 multipliers. The proposed filter occupies only 41% of the area of the reference filter. The power consumption for the digital filter with exact multipliers is 63% higher than the power consumption for the proposed digital filter with AO-RAD4 approximate multipliers. The proposed digital filter consumes 70% less energy (PDP) than the digital filter with exact multiplication. Besides, the proposed digital filter can process the samples 1.2 time faster.

The superior hardware performance of the proposed approximate filter originates from the usage of the approximate multipliers. The proposed approximate filter and the reference filter have the same FOS structure and the same number of arithmetic operations. Still, the former employs the approximate AO-RAD4 multipliers, and the latter employs the exact radix-4 multipliers. In this way, we achieved fair comparison and eliminated the influence of the filter structure on the synthesis results. For the exact radix-4 multiplier, the complexity of the product generation stage equals $O(n^{2}/2)$ (*n* bits of multiplicand X, and n/2 partial products). In the case of the AO-RAD4 approximate multiplier, the complexity is equal to O(n·(q−M)/2), where *n* denotes the bit width of the multiplicand, *q* quantization factor, and *M* represents the truncation parameter of approximate odd radix-4 Booth multiplier. In the proposed filter, we chose n=32 bits for representing the multiplicand X, q=10 for the quantization factor, and employed AO-RAD4 multipliers with M=4 and M=5. Therefore, the partial product stage complexity in AO-RAD4 is theoretically reduced by 80% compared to the exact radix-4 multiplier. The exact multiplier and the proposed AO-RAD4 multiplier also differ in the number of partial products. The exact radix-4 Booth multiplier has n/2 partial products, and the proposed approximate multiplier has (q−M)/2 partial products. The employed approximate multipliers with M=4 and M=5 have only three partial products, and the exact multiplier has 16 partial products. With fewer partial products, the approximate AO-RAD4 multiplier exhibits significantly smaller energy consumption and area utilization, which leads to an overall reduction in area and energy in the proposed filter.

To compare the Verilog model and MATLAB model outputs, we conducted the verification through FPGA prototyping. We deployed the proposed filter to the Zync 7000 SoC on the ZYBO Z7 FPGA development board. For the test inputs, we used impulse sequence and Gaussian white noise (AWGN). Figure 13 shows the filter’s outputs from the filter implemented in Zync 7000 SoC in the presence of the environmental noise and MATLAB simulation model. We can notice that the outputs match, and the filter implemented in Zync 7000 SoC has the same functionality as the MATLAB model.

### 6.4. Discussion

Employment of approximate multipliers in the A-weighting IIR filter offers remarkable savings in energy consumption and area utilization, and it has a negligible impact on its accuracy. As the approximate and the reference filter have the same structure, and the same number of first-order sections, the low area utilization and low energy consumption in the approximate filter comes solely from the employment of the approximate multipliers. The smaller number of partial products in the proposed approximate multiplier leads to a smaller circuit. Hence, the overall area and power consumption of the proposed filter have been reduced. However, careful placement of approximate multipliers in the A-weighting filter is required to meet the A-weighting filter’s accuracy, stability, and frequency response. As the criteria for placement of the approximate multipliers, we selected the similarity between magnitude responses of the proposed filter and reference filter. In other words, the optimal choice and placement of the approximate multipliers in the A-weighting filter give the magnitude response, which is almost identical to the magnitude response of the referenced A-weighting filter and satisfies the IEC 61672-1 requirements. Hence, there is an insignificant difference between signals filtered with the proposed and reference filters.

As with every approximation scheme, the one proposed here also has shortcomings and limitations. The proposed approximation scheme applies only to the IIR filters that can be implemented through decomposition on the first-order sections (FOS). We selected the first-order sections as a filter building block because they have a linear relationship between coefficients and poles of a transfer function. On the other hand, we can decompose on FOS only the IIR filters with real poles or near-real poles. Besides, this study solely concentrated on deploying approximate multipliers, and the design of adders was unaltered. To further improve the proposed filters’ power consumption, we need to consider the adders’ design. To summarize, Figure 14 shows the design flow presented in this paper.

## 7. Conclusions

In this paper we proposed an energy-efficient A-weighting IIR digital filter that uses approximate multiplications and coefficient quantization. We have thoroughly assessed the impacts of quantization, pole-zero pairings, the positions of the first-order sections in the filter’s cascade, and the placement of AO-RAD4 approximate multipliers in the filter’s cascade on its performance. The proposed A-weighting IIR digital filter has an almost identical frequency response to the filter with exact multipliers while consuming around 70% less energy. Experiments on acoustic noise suggest that the proposed digital A-weighting filter can be deployed in environmental noise measurement applications without any notable performance degradation. In future work, we will tackle the challenges of employing approximate arithmetic in second-order sections and extending the proposed approach to general digital IIR filter design. Further research will concentrate on the employment of error correction circuits and lowering the error caused by truncation in fixed-point arithmetic. 

## Figures and Tables

**Figure 1 sensors-21-00732-f001:**
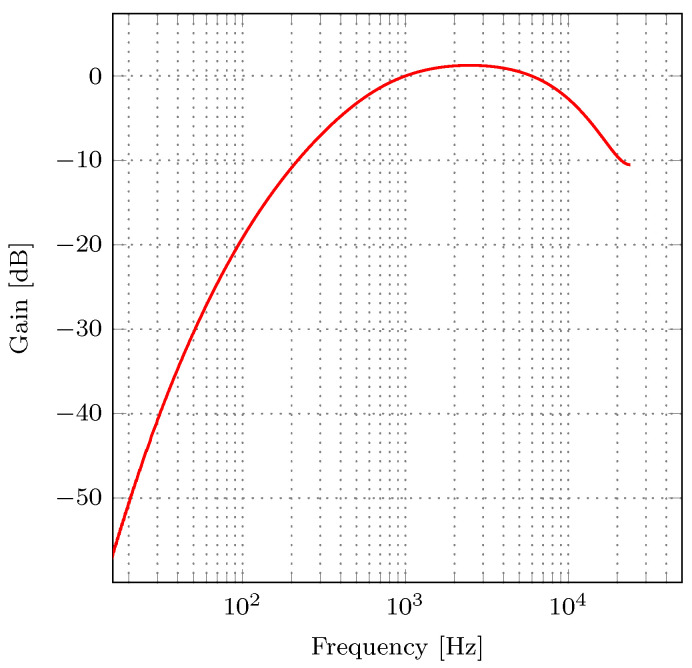
Magnitude response of the analog A-weighting filter given by (Equation 3).

**Figure 2 sensors-21-00732-f002:**
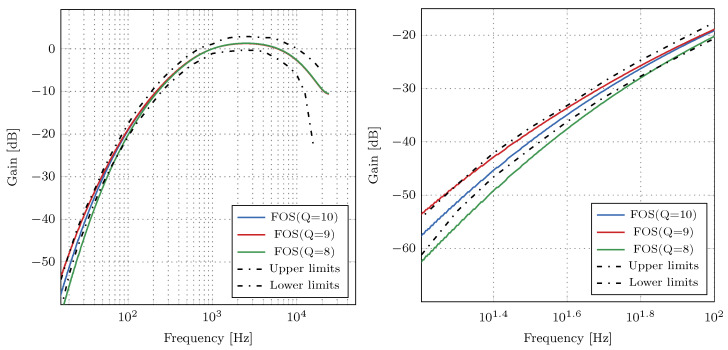
Magnitude responses of the digital A-weighting filter for various values of *Q*. **Left**: magnitude responses; **right**: enlarged portion of the magnitude responses. The filter with Q=10 satisfies the tolerance limits for the A-weighting filter.

**Figure 3 sensors-21-00732-f003:**
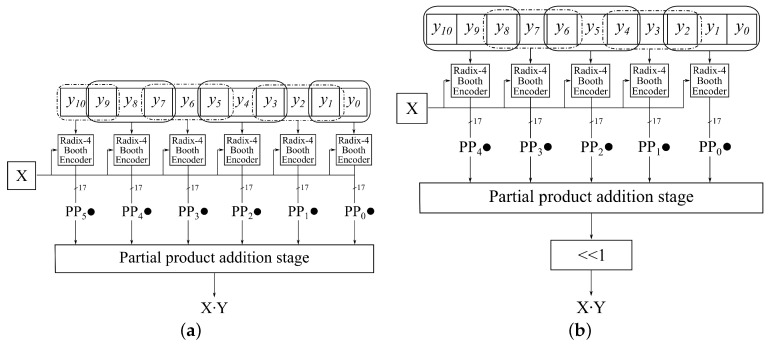
Exact and approximate odd radix-4 multiplier. (**a**) Exact radix-4 multiplier; (**b**) Approximate odd radix-4 multiplier.

**Figure 4 sensors-21-00732-f004:**
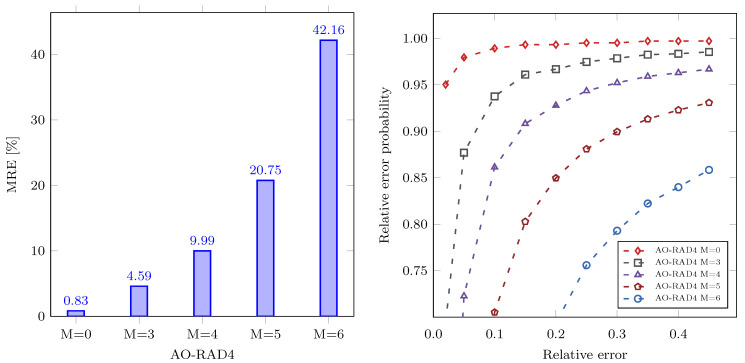
Error analysis of an approximate odd radix-4 (AO-RAD4) multiplier for different values of parameter M. **Left**: mean relative error (MRE); **right**: the probability that the relative error is smaller than a specific value.

**Figure 5 sensors-21-00732-f005:**
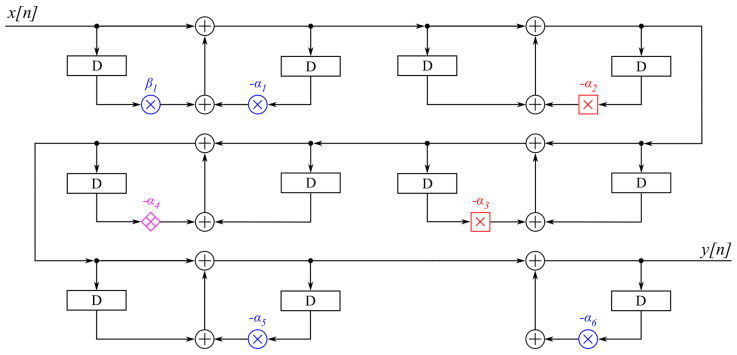
The proposed A-weighting filter. The 
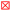
 denotes the exact radix-4 multiplier, 
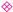
 denotes AO-RAD4 with M = 0, and 
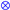
 denotes AO-RAD4 with M = 5. $\alpha_{j}$ and $\beta_{j}$ represent coefficients $\alpha_{1}$ and $\beta_{1}$ of the *j*-th first-order sections (FOS).

**Figure 6 sensors-21-00732-f006:**
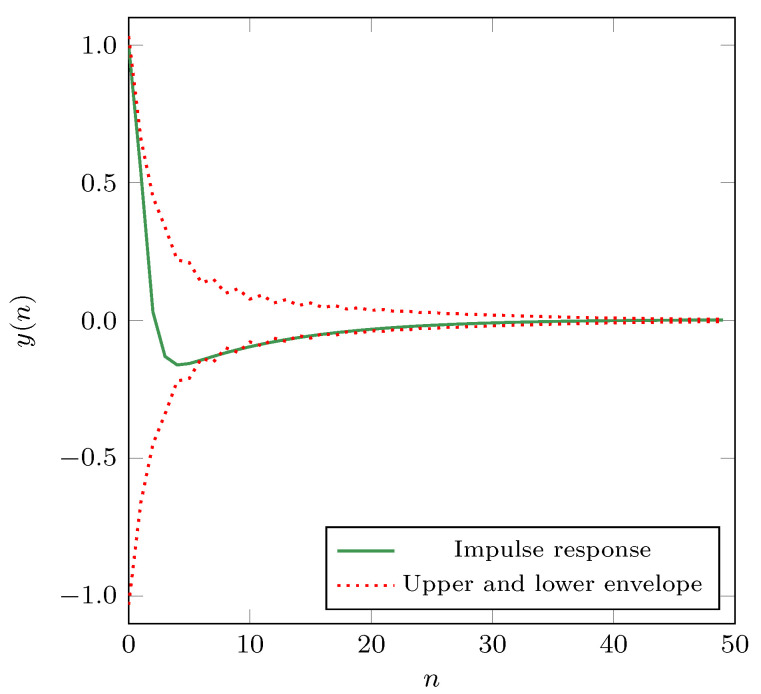
The impulse response of the proposed filter and its envelopes.

**Figure 7 sensors-21-00732-f007:**
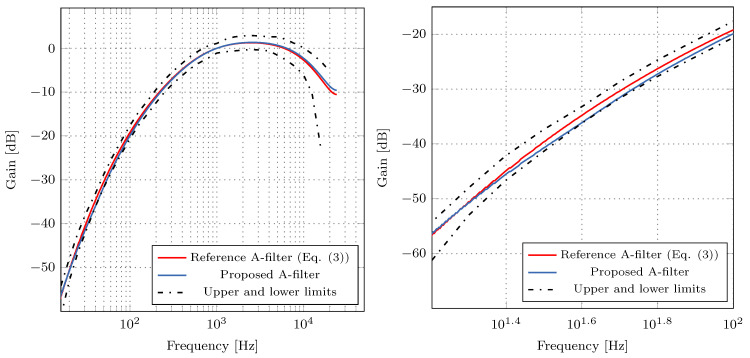
Magnitude responses of the proposed and reference digital A-weighting filters. **Left**: Magnitude response; **Right**: enlarged portion of the magnitude response.

**Figure 8 sensors-21-00732-f008:**
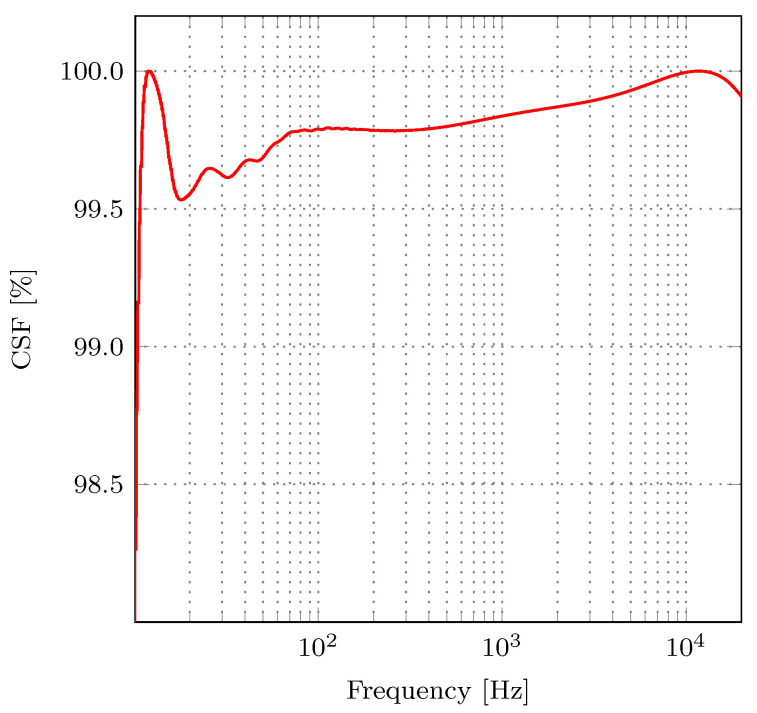
Cross signature scale factor (CSF) of frequency responses of the proposed and reference A-weighting filters.

**Figure 9 sensors-21-00732-f009:**
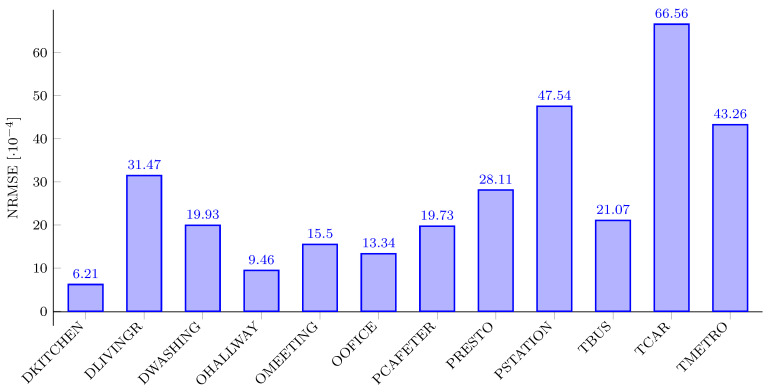
NRMSE between the signal from the reference filter and the signal from the proposed filter for different recordings in the DEMAND database.

**Figure 10 sensors-21-00732-f010:**
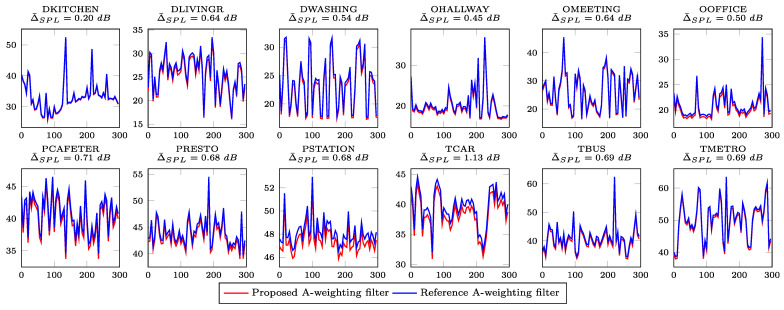
Sound pressure level profile for each of the recordings in the DEMAND collection. ${\bar{\Delta }}_{SPL}$ denotes the mean absolute error between the SPL values obtained with the proposed and the reference A-weighting filters.

**Figure 11 sensors-21-00732-f011:**
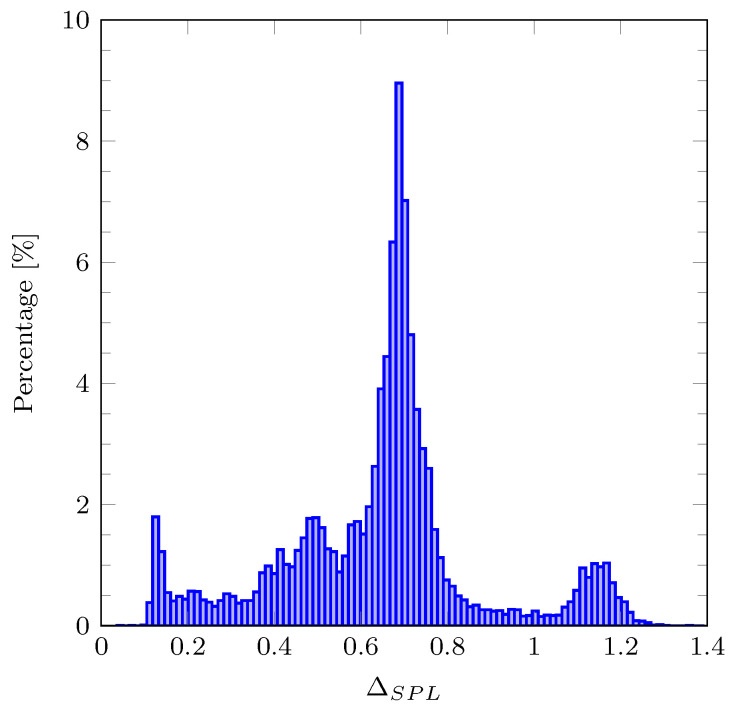
Histogram of the $\Delta_{SPL}$ values from the DEMAND dataset. $\Delta_{SPL}$ denotes the absolute error between the SPL values obtained with the proposed approximate and the reference A-weighting filters.

**Figure 12 sensors-21-00732-f012:**
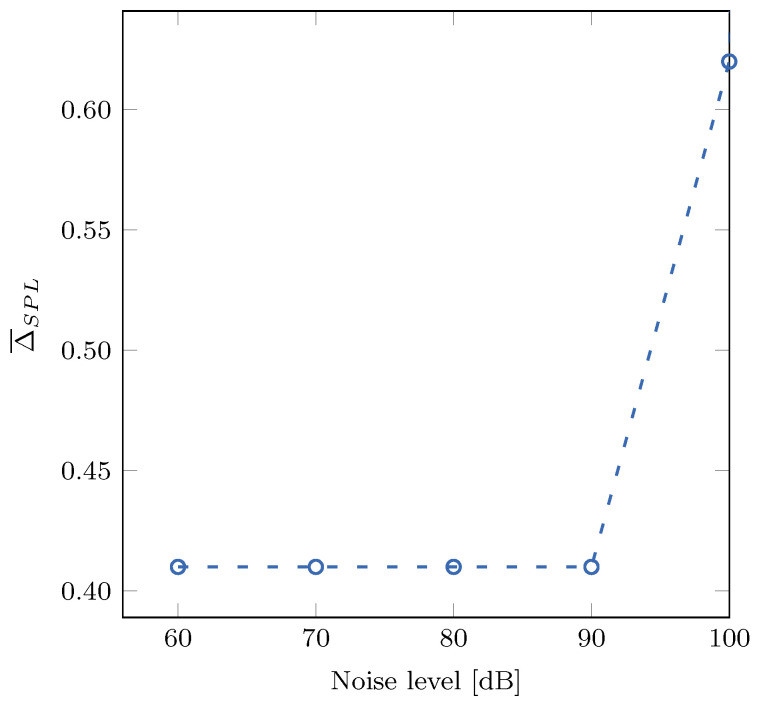
Correlation between the mean absolute error ${\bar{\Delta }}_{SPL}$ and the pink noise level.

**Figure 13 sensors-21-00732-f013:**
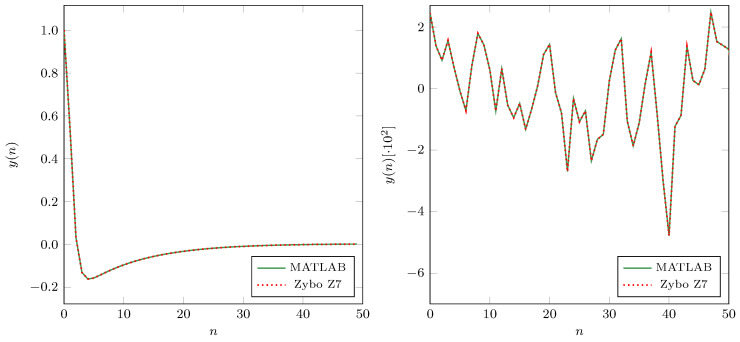
Filter’s outputs from the Zync 7000 SoC and of the MATLAB model for two different inputs. **Left**: response to the impulse signal; **Right**: response to AWGN.

**Figure 14 sensors-21-00732-f014:**
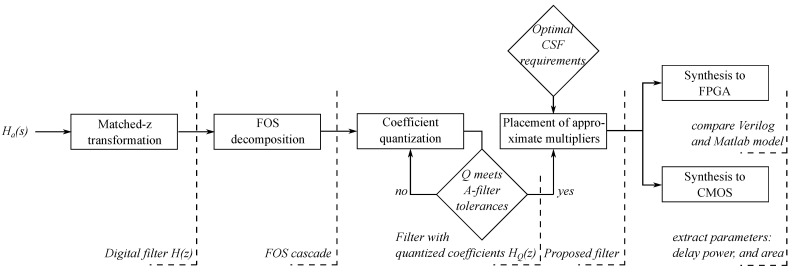
The proposed design flow.

**Table 1 sensors-21-00732-t001:** Cross signature scale factor (CSF) for different parameter M values: combinations that satisfy tolerance for A-weighting filtering are marked in green, otherwise in red.

		AO-RAD4				
		*M* = 0	*M* = 3	*M* = 4	*M* = 5	*M* = 6
**Factors**	**207**	100.00	99.99	99.95	99.95	99.95
	**307**	100.00	100.00	100.00	99.99	99.93
	**932**	100.00	99.95	99.95	99.95	97.35
	**1010**	100.00	99.77	99.77	92.36	79.26
	**1021**	98.86	88.16	76.62	66.74	57.64

**Table 2 sensors-21-00732-t002:** Synthesis results.

Filter	Area [μm^2^]	Delay [ns]	Power [μW]	PDP [fWs]
Proposed approximate	7588.71	1.97	203.75	401.39
Exact 32 × 32	18,518.92	2.39	526.19	1257.59

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
