# Peer review of "On the Design of an Energy Efficient Digital IIR A-Weighting Filter Using Approximate Multiplication"

_sensors, 2021, doi:10.3390/s21030732_

Round 1

Reviewer 1 Report

The paper is carefully written and well organized. The language presented in it is smooth and easily understandable which makes reading it a nice experience.

The introduction and background sections are quite comprehensive and a satisfactory number of citations can be found in them. Subsequent sections thoroughly explain the whole way of thinking behind the proposed A-weighted filter design, and finally, simulation experiments are carried out adequately.

I especially like the graphical aspects of the presentation of concepts and results. Figures are self-explanatory, and the authors managed to clearly emphasize the most interesting features. 

There is one thing which I would like to be better explained: In lines no. 191-192 authors claim that reference filter (3) is assumed to be represented with exact coefficient values (infinite number of bits). How this is achieved in simulations using MATLAB which uses a finite number of bits to represent numbers?

I also think that I found a minor error in the legend of Fig. 7. I think that the reference filter is defined in Eq. 3 (not Eq.7).   

Author Response

Best regards

Ratko Pilipović, Vladimir Risojević and Patricio Bulić

Reviewer 2 Report

This paper presents a new idea to design digital IIR A-weighting filter using approximate multiplication. Detail description about the method has given. Also, simulation and experimental results verify the authors' finding.

The reviewer only suggests the authors can prepare a flow chart or show their designing processes steps by steps. I believe this will be helpful for the readers to realize your contribution and implement your work. By the way, the way to extend your work to general digital IIR filter design, say SOS, can be included too.

Other comments:

  1. There is typo in Eq.3 or Eq. 5 (H2). Please double check.
  2. A low pass filter is add to Eq.3, can the author explain why? Also, how to decide the cutoff frequency?

Author Response

(The authors gave the same response as above.)
